CSQUiD: an index and non-probability framework for constrained skyline query processing over uncertain data

http://orcid.org/0000-0003-3672-7968 Lawal Ma'aruf Mohammed 1
Ibrahim Hamidah 2 hamidah.ibrahim@upm.edu.my
Mohd Sani Nor Fazlida 2
Yaakob Razali 2
Alwan Ali A. 3
1 Department of Computer Sciences, Faculty of Physical Sciences, Ahmadu Bello University , Zaria, Kaduna , Nigeria
2 Department of Computer Science, Faculty of Computer Science and Information Technology, Universiti Putra Malaysia , Serdang, Selangor , Malaysia
3 School of Theoretical and Applied Science, Ramapo College of New Jersey , Mahwah, New Jersey , United States
Pires Ivan Miguel
Electronic publication date: 2024 Sep 16
Publication date: 2024
Volume: 10
Electronic Location ID: e2225
Received 2023 Aug 24; Accepted 2024 Jul 10
Copyright: © 2024 Lawal et al.
Copyright year: 2024
Copyright holder: Lawal et al.
License: This is an open access article distributed under the terms of the Creative Commons Attribution License, which permits unrestricted use, distribution, reproduction and adaptation in any medium and for any purpose provided that it is properly attributed. For attribution, the original author(s), title, publication source (PeerJ Computer Science) and either DOI or URL of the article must be cited.
License URL: https://creativecommons.org/licenses/by/4.0/

Keywords: Skyline, Uncertain data, Constrained skyline query, MBR

Funding: Ministry of Higher Education Malaysia under the Fundamental Research Grant Scheme FRGS/1/2020/ICT03/UPM/01/1 Universiti Putra Malaysia This work was supported by the Ministry of Higher Education Malaysia under the Fundamental Research Grant Scheme (FRGS/1/2020/ICT03/UPM/01/1) and the Universiti Putra Malaysia. The funders had no role in study design, data collection and analysis, decision to publish, or preparation of the manuscript.

==============================
Uncertainty of data, the degree to which data are inaccurate, imprecise, untrusted, and undetermined, is inherent in many contemporary database applications, and numerous research endeavours have been devoted to efficiently answer skyline queries over uncertain data. The literature discussed two different methods that could be used to handle the data uncertainty in which objects having continuous range values. The first method employs a probability-based approach, while the second assumes that the uncertain values are represented by their median values. Nevertheless, neither of these methods seem to be suitable for the modern high-dimensional uncertain databases due to the following reasons. The first method requires an intensive probability calculations while the second is impractical. Therefore, this work introduces an index, non-probability framework named Constrained Skyline Query processing on Uncertain Data (CSQUiD) aiming at reducing the computational time in processing constrained skyline queries over uncertain high-dimensional data. Given a collection of objects with uncertain data, the CSQUiD framework constructs the minimum bounding rectangles (MBRs) by employing the X-tree indexing structure. Instead of scanning the whole collection of objects, only objects within the dominant MBRs are analyzed in determining the final skylines. In addition, CSQUiD makes use of the Fuzzification approach where the exact value of each continuous range value of those dominant MBRs’ objects is identified. The proposed CSQUiD framework is validated using real and synthetic data sets through extensive experimentations. Based on the performance analysis conducted, by varying the sizes of the constrained query, the CSQUiD framework outperformed the most recent methods (CIS algorithm and SkyQUD-T framework) with an average improvement of 44.07% and 57.15% with regards to the number of pairwise comparisons, while the average improvement of CPU processing time over CIS and SkyQUD-T stood at 27.17% and 18.62%, respectively.

Introduction

The continuous increase in the deployment of modern solutions that make use of database applications, such as online reservation systems for hotel, airline, and business transactions; the deployment of e-health support system, fog-cloud computing, and health edge computing; employment of workflow scheduling applications for handling multi-objective optimization problem, and hosting of other portals to name a few has resulted in the capture or generation of massive volume and varieties of data. With the advancement in computing, uncertain data widely exist in many database applications. Due to the immense volume of these uncertain data, accessing data for supporting many practical applications becomes tedious (Khalefa, Mokbel & Levandoski, 2010; Kuo et al., 2022; Li et al., 2012, 2017, 2019; Lian & Chen, 2013a; Liu et al., 2013; Qi et al., 2010; Saad et al., 2014, 2016, 2018, 2019). This study defines uncertain data as those with continuous range values where their exact values are not explicitly represented in the database.

In most real-world database applications, a database represents requisite information in a variety of ways. For instance, several accommodation portals exist, like Rent.com (https://www.rent.com), Apartment.com (https://www.apartments.com/), ForRent.com (https://www.forrent.com/), Rentals.com (https://www.rentals.com/), Gottarent.com (https://www.gottarent.com), and showcase.com (https://www.showcase.com). Often times, information retrieved by these platforms contains objects with different representations of data values which undoubtedly influenced how queries are processed. Figure 1 presents a snapshot of the showcase.com website for New York properties in the United States of America (https://www.showcase.com). The website contains the following dimensions: Name of Property, Size, Rent Rate, Office Space, and Deposit. From this snapshot, examples of continuous range values are 2,598–37,143, 10,024–30,905, 3,906–8,832, 1,891–18,241, and 1,464–6,559 as presented under the Size dimension. These values reflect uncertain data as their precise values are not known.

Figure 1 Database for properties at New York in USA.

The skyline operator introduced by Borzsonyi, Kossmann & Stocker (2001) plays an important role to accurately and efficiently solve problems that involve user preferences. The skyline operator is used to identify dominant objects also known as skyline objects from a potentially large multi-dimensional collection of objects by keeping only those objects that are not worse than any other. The dominant objects are said to be the best, most preferred set of objects; hence fulfilling the user preferences. Enormous number of skyline algorithms have been proposed since its introduction two decades ago, dealing with a wide variety of queries including skyline queries (Afshani et al., 2011; Alwan et al., 2016; Atallah & Qi, 2009; Bartolini, Ciaccia & Patella, 2006, 2008; Chomicki et al., 2005; Han et al., 2013; Jiang et al., 2012; Khalefa, Mokbel & Levandoski, 2010; Kossmann, Ramsak & Rost, 2002; Lawal et al., 2020b; Mohamud et al., 2023; Liu et al., 2013; Papadias et al., 2003, 2005; Pei et al., 2007; Saad et al., 2014, 2016, 2018, 2019; Tan, Eng & Ooi, 2001; Wang et al., 2009), range query (Saad et al., 2016, 2019), reverse skyline query (Gao et al., 2014; Lim et al., 2016; Xin, Bai & Wang, 2011), etc.

Unlike skyline query that requires the whole collection of objects to be analyzed during dominance testing, processing a constrained skyline query requires the collection of objects to be filtered before the dominance testing is performed. Only those objects that lie within the query are collected for dominance testing while the rest are considered as not satisfying the constraint specified in the query. Intuitively, identifying whether an object satisfies a constrained skyline query is straightforward when dealing with certain data. Skyline computations are typically performed on objects whose exact values are within the constraints specified in the constrained skyline query. Likewise, objects with uncertain data require additional evaluation which increases the complexities of evaluating the constrained skyline query.

Although there are quite a number of works on skyline queries over uncertain data (Atallah & Qi, 2009; Jiang et al., 2012; Khalefa, Mokbel & Levandoski, 2010; Li et al., 2017, 2019; Pei et al., 2007; Saad et al., 2014, 2016, 2018, 2019; Zhou et al., 2015) only a few that focus on constrained skyline queries (Papadias et al., 2005; Qi et al., 2010). These works either employ a probability-based approach that requires additional computational cost due to intensive probability calculations (Atallah & Qi, 2009; Godfrey, Shipley & Gryz, 2005; Jiang et al., 2012; Khalefa, Mokbel & Levandoski, 2010; Lian & Chen, 2013a, 2013b; Lim et al., 2016; Liu & Tang, 2015; Pei et al., 2007; Saad et al., 2016, 2019) or assume that the uncertain values can be represented by their median values (Li et al., 2012) which is impractical for many modern database applications in which the rate of data uncertainty is reasonably high. To efficiently address the challenges associated with computing constrained skyline queries over uncertain data, in this article we proposed an index, non-probability framework named Constrained Skyline Query processing on Uncertain Data (CSQUiD). Given a collection of objects with uncertain data, the CSQUiD framework constructs the minimum bounding rectangles (MBRs) by employing the X-tree indexing structure. The idea of the proposed method relies on accumulating objects with similar uncertainty values within the same MBR. This intuitive process guarantees that the computational cost of skyline computation is significantly minimized by analyzing the associations between MBRs and between MBRs and the constrained skyline query. Instead of identifying those objects whose dimension values are within the constraints of a constrained skyline query as proposed in CIS algorithm (Li et al., 2012), these objects are identified based on the MBRs that lie within the constraints of the query. By doing so, irrelevant objects are discarded as early as possible. This leads to avoiding many unwanted pairwise comparisons between objects, which in turn results in a significant reduction in the processing time of the skyline process. Moreover, to deal with the issue of uncertainty of data, the CSQUiD makes use of the Fuzzification approach where the exact value of each continuous range value of those dominant MBRs’ objects is identified. This will provide a significant advantage to our proposed solution by allowing the conventional skyline algorithm to be easily employed for computing the final skylines.

The following points summarize the main contributions of this article: i) We have proposed an index, non-probability framework, named Constrained Skyline Query processing on Uncertain Data (CSQUiD) designed to tackle the issue of processing constrained skyline queries over uncertain data.

ii) We have formally extended the concept of object dominance to MBRs and introduced a variant of dominance relationship, named Dominance Relationship between MBRs, which identifies the skyline objects based on the identified dominant MBRs.

iii) We have conducted extensive experiments to demonstrate the CSQUiD’s superiority over the most recent solutions in processing constrained skyline queries and ultimately identifying the dominant objects over uncertain data.

This article is organized as follows: The ‘Related Work’ section reviews the methods proposed by previous studies that are related to the work presented in this article. The Preliminary section introduces the notations and the terms that are frequently used throughout the article. This is followed by the ‘Proposed Framework’ section which presents the proposed framework, CSQUiD. Meanwhile, the ‘Result and Discussion’ section evaluates the performance of the proposed framework and compares the results to other previous works. The last section which is the ‘Conclusion’ section, concludes this work and sheds light on some directions which can be pursued in the future.

Motivating example

Uncertainty of data is inherent in many emerging applications such as sensor networks, data integration and cleaning, record lineage, spatio-temporal and scientific data management, and so forth, and query processing including skyline query over uncertain data has gained widespread attention. Nonetheless, the existing skyline algorithms assume that the query requirements of the users are based on the same fixed set of dimensions (all dimensions) that are available in the data set and users are assumed to be only interested in either the minimum or maximum value over the whole space of each given dimension. This rigid assumption often lead to an impractical skyline query which no longer offer any interesting insights. In practice, different users may be interested in different range of values (subspace) of a given dimension. For instance, a user may only be interested in those apartments whose price rates are between $50–$60. In this regard, the constrained skyline queries where constraints on specific dimensions are being attached to the query requirements give flexibility to users in specifying their interests. Nonetheless, deriving skylines of a constrained skyline query over uncertain data is challenging. This is illustrated through the examples given in Fig. 2.

Figure 2 (A) Skyline query with certain data (B) Skyline query with uncertain data (C) Constrained skyline query with certain data (D) Constrained skyline query with uncertain data (E) Constrained skyline query with uncertain data and MBR.

Figure 2A presents the set of skylines, S, that is derived based on the given set of certain data, D = {A, B, C, D, E, F}. With the assumption that apartments with the lowest price (minimum price) and nearest to the beach (minimum distance) are preferrable, then S = {C, E, A} while objects B, D, and F are not the skyline objects. Meanwhile, Fig. 2B shows the skyline result of a given set of uncertain data. For simplicity, the uncertain value is reflected by object F over the price dimension. In this example, S = {C, E, A} and the object C dominates the object F regardless the point where the exact value of F will fall along the range $50–$100. Nonetheless, an object with uncertain value may have chances to dominate other objects. This is further discussed in the example given in Fig. 2D.

On the other hand, Fig. 2C shows the result of S when a constrained skyline query is specified over a set of certain data. The objects that fall within the specified constraint, i.e., price between $75 and $125, are D, E, and F while the set of skylines, S = {E, F}, since both objects E and F dominate D in both dimensions. However, identifying skylines of a constrained skyline query when uncertain data are inevitable is not straightforward. The main challenges are as explained below: (i) the conventional dominance relationship as defined in Definition 1 works by comparing the values of each dimension of the objects being analysed in determining the dominant objects. The comparison is straightforward if the values to be compared are precise (certain) values. However, the dominance relationship cannot be applied directly to compare a precise value/continuous range value against another continuous range value. For instance, consider F=(3,[50−100]) and E=(1.5,100), we cannot certainly conclude that [50−100]≤100 without knowing the exact value of F.d2. (ii) A constrained skyline query contains constraints on specific dimensions. If the value of an object on the dimension being constrained falls within the specified constraints, then the object is said to be relevant to the given query. However, to decide if an object falls within the specified constraints is not straightforward when the continuous range values of the objects intersect with either the lower bound and/or upper bound of the constrained skyline query. This is shown in Fig. 2D where the price value of object F is in the range of $50–$100. Using the same constrained skyline query, the object F is said to fall within the constraint of the query if its price value falls in the range $75–$100 and otherwise if its price value is less than $75. Moreover, we cannot say for sure that the object E is better than F and vice versa. If the exact price value of F is less than $100, then both E and F do not dominate each other; while F is dominated by E if its price value is equal to 100. The work by Li et al. (2012) assumes that the exact value of a continuous range value is represented by its median value. Hence, the price value of object F is always $75 (i.e., (50 + 100)/2) which results in the skyline set, S = {E, F}. This approach is not realistic since not all continuous range values will fall in the median point, especially objects/values with seasonal effect. On the other hand, by employing the probability-based approach as used by Atallah & Qi (2009), Coffman-Wolph (2016), Gao et al. (2014), Li et al. (2012, 2014), Lim et al. (2016), Papadias et al. (2005), Qi et al. (2010), Saad et al. (2014, 2016, 2018, 2019), the probability that object F dominates E is 49%. This is calculated as follows: Pra(F<E)=∫75100−0.5f(F)dF=150(99.5)−150(75) = 1.99 − 1.5 = 0.49. Nevertheless, F will be the skyline object if its probability value is greater or equal to a threshold value, τ, set by the user. If τ = 75%, then F will not be listed as one of the skyline objects. This approach is prohibitively expensive as calculation against every object that has the potential to dominate an object needs to be performed including objects having low chances of being the skyline objects. Also, if the τ value is high, then the number of skyline objects derived is small while a low value of τ will result in a huge number of skyline objects.

In contrast to the previous approaches, we employed the following approaches (i) X-tree indexing technique where objects are organized into MBRs and (ii) Fuzzification approach where the exact value of each continuous range value is determined. Figure 2E presents the MBRs that are derived based on the given set of data. Two MBRs are constructed, namely: R1 and R2. Based on the given constrained skyline query and dominance relationship between MBRs (see Definition 9), it is clear that the MBR R1 dominates the MBR R2. This will disregard object D from being considered in skyline computation at an earlier stage. Then, by employing the Fuzzification approach, the exact price value of F is identified. If the predicted value is less than 75, then F will not be considered in the skyline computation, otherwise if its value is less than 100, then the set of skyline objects, S = {E, F}. In our approach, the exact value of a continuous range value is not necessarily the median value; while extensive probability computation and thresholding are simply avoided.

Related works

After the introduction of the skyline operator by Borzsonyi, Kossmann & Stocker (2001), many variations of skyline algorithms have been proposed. Among the earlier and notable skyline algorithms reported in the literature are Skyline Sorted Positional List (SSPL) (Han et al., 2013), Bitmap (Tan, Eng & Ooi, 2001), Nearest—Neighbor (NN) (Kossmann, Ramsak & Rost, 2002), Divide-and-Conquer (D&C) (Borzsonyi, Kossmann & Stocker, 2001), Branch and Bound Skyline (BBS) (Papadias et al., 2005), Block-Nested-Loop (BNL) (Borzsonyi, Kossmann & Stocker, 2001), Sorted Filter Skyline (SFS) (Chomicki, 2003), Linear Eliminate Sort for Skyline (LESS) (Godfrey, Shipley & Gryz, 2005), and Sort and Limit Skyline (SaLSa) (Bartolini, Ciaccia & Patella, 2008). Although there are several works reported in the literature that focus on skyline analysis on uncertain data, they differ with regard to the type of uncertain data being handled. Works like Atallah & Qi (2009), Jiang et al. (2012), Liu & Tang (2015), and Pei et al. (2007), developed solutions for data with discrete uncertainty; while Khalefa, Mokbel & Levandoski (2010), Li et al. (2017, 2019), and Saad et al. (2014, 2016, 2018, 2019) assumed uncertain objects that are represented as an interval value. Most of the reported works like Khalefa, Mokbel & Levandoski (2010), Li et al. (2017, 2019), and Saad et al. (2014, 2016, 2018, 2019) rely on the probabilistic skyline model to compute the probability of an uncertain object to be in the skyline. There are also works that focus on a specific environment/platform like distributed database (Li et al., 2017; Zhou et al., 2015). With regard to the type of queries handled by these works, Papadias et al. (2005) and Qi et al. (2010) focus on range query, Li et al. (2019) on parallel k-dominant skyline queries, Lian & Chen (2009) on top-k dominating queries, while most of the previous works like Khalefa, Mokbel & Levandoski (2010), Li et al. (2017, 2019), and Saad et al. (2014, 2016, 2018, 2019) emphasized on skyline queries. In the following, we report some of these works that are relevant to the work presented in this article.

A novel probabilistic skyline model is introduced by Pei et al. (2007) to tackle the problem of skyline analysis on uncertain data. In this work, probabilistic skylines also known as p-skylines are identified as those uncertain objects whose skyline probabilities are at least p. To derive p-skylines over large uncertain data, the top-down and bottom-up algorithms are developed. Using a selection of instances of uncertain objects and their computed skyline probabilities, the bottom-up algorithm effectively prunes other instances and uncertain objects. Meanwhile, the top-down algorithm aggressively prunes both subsets and objects after recursively partitioning the instances of uncertain objects into subsets.

With the assumption that low probability events cannot be simply ignored, the work in (Atallah & Qi, 2009) studied the problem of computing skyline probabilities for data with discrete uncertainty; hence eliminating thresholding. An efficient algorithm based on space partitioning and weighted dominance counting is proposed. The skyline analysis is done only once for all users. By returning skyline probabilities of all instances provides flexibility to the users in identifying their own interesting skyline instances.

The problem of computing the probability of a point with an uncertain location lying on the skyline is investigated by Afshani et al. (2011). Two algorithms are presented to exactly compute the probability that each uncertain point, described as a probability distribution over a discrete set of locations, is on the skyline; while two new near-linear time algorithms were introduced for approximately computing the probability that each uncertain point is on the skyline.

Li et al. (2012) first identified the issue of skyline query on uncertain data, where each dimension of the uncertain object is represented as an interval or an exact value. Two efficient algorithms are devised, namely: Branch and Bound Interval Skyline (BBIS) and Constrained Interval Skyline (CIS) with I/O optimal for the conventional interval skyline queries and constrained interval skyline queries, respectively.

Meanwhile, Jiang et al. (2012) examined the problem of skyline analysis with uncertain data. Similar to the idea of the work in Pei et al. (2007), a novel probabilistic skyline model is proposed where an uncertain object may take a probability to be in the skyline, and a p-skyline contains all objects whose skyline probabilities are at least p ( 0<p≤1). A bounding-pruning-refining framework and three algorithms were systematically developed. These algorithms are named bottom-up, top-down, and hybrid algorithms. Both the top-down and bottom-up algorithms are as suggested by Pei et al. (2007), while the hybrid algorithm is presented to combine the benefits of the first two algorithms to further enhance performance.

To resolve the issue of setting a probabilistic threshold to qualify each skyline tuple independently as developed by p-skyline, Liu et al. (2013) propose a new uncertain skyline query, called U-Skyline query. U-Skyline query searches for a set of tuples that has the highest probability (aggregated from all possible scenarios) as the skyline answer. In order to answer the U-Skyline queries efficiently, a number of optimization techniques were introduced that are: computational simplification of U-Skyline probability, pruning of unqualified candidate skylines and early termination of query processing, reduction of the input data set, and partition and conquest of the reduced data set.

In Li et al. (2017), a distributed skyline query is defined to address the skyline queries over interval data, which is a special kind of attribute-level uncertain data that widely exists in many contemporary database applications. Two efficient algorithms were devised to retrieve the skylines progressively from distributed local sites with a highly optimized feedback framework. To further improve the queries, two strategies are exploited.

Similar to the work of Pei et al. (2007), Saad et al. (2014, 2018) proposed a framework named SkyQUD to efficiently answer skyline queries on high dimensional uncertain data with objects represented as continuous ranges and exact values, which is referred to as uncertain dimensions. The term is introduced to emphasize that a particular dimension may contain both continuous ranges and exact values. The skyline objects are determined through three methods that guaranteed the probability of each object being in the final skyline results. These methods are exact domination, range domination, and uncertain domination which follows the filter-refine approach. The SkyQUD framework is designed to be able to accept a threshold value that is specified by users, in which each object must exceed to be recognized as a dominant object.

Utilizing similar concept, Saad et al. (2016, 2019) extended their solution reported in Saad et al. (2014) to compute skyline with range query issued on uncertain dimensions. The proposed framework named SkyQUD-T eliminates objects that do not satisfy a given query range before advance processing is performed on the surviving objects. The work attempts to support users that would query information in a range of search rather than a fixed search.

Table 1 summarizes the research works presented in this section that mainly focus on uncertain data. It presents the reference, the approach employed, the proposed algorithm(s), type of query, type of data, and the data sets that these works have utilized in their performance analysis. From this summary, the following can be concluded: (i) Most of these works focus on skyline query with the assumption that users are only interested in either the minimum or maximum value over the whole space of each given dimension (Afshani et al., 2011; Atallah & Qi, 2009; Jiang et al., 2012; Li et al., 2017; Liu et al., 2013; Pei et al., 2007; Saad et al., 2014, 2018). Only a few studies like Li et al. (2012) and Saad et al. (2016, 2019) focus on constrained skyline query (also known as interval and range query) (ii) There are two types of data uncertainty being explored, namely: discrete uncertainty (object is associated with multiple instances) (Afshani et al., 2011; Atallah & Qi, 2009; Jiang et al., 2012; Liu et al., 2013; Pei et al., 2007) and continuous uncertainty (also known as interval and range) (Li et al., 2012, 2017; Saad et al., 2014, 2016, 2018, 2019). Hence, the closest works that can be compared to our work which mainly focus on constrained skyline query with continuous uncertainty are the CIS algorithm (Li et al., 2012) and SkyQUD-T (Saad et al., 2016, 2019).

Table 1 Summary of related works on uncertain data.

Reference	Approach	Algorithm	Type of query	Type of data	Data set	Limitation	
Pei et al. (2007)	Probabilistic skyline	Bottom-up, Top-down	Probabilistic skyline	Discrete uncertainty	NBA, synthetic	Require additional computational cost due to intensive probability calculation	
Atallah & Qi (2009)	Probabilistic skyline	Sub-Quadratic based on space partitioning and weighted dominance counting	Probabilistic skyline	Discrete uncertainty	Synthetic	Require additional computational cost due to intensive probability calculation	
Liu et al. (2013)	Probabilistic skyline	Dynamic Programming Framework	U-Skyline	Discrete uncertainty	Used Cars, synthetic	Rigid assumptions where users are only interested in either the minimum or maximum value over the whole space of a given dimension	
Afshani et al. (2011)	Probabilistic skyline	Deterministic and Monte Carlo algorithms	Skyline	Discrete uncertainty	–	Rigid assumptions where users are only interested in either the minimum or maximum value over the whole space of a given dimension	
Li et al. (2012)	R*-tree and Median approach	Branch-and-Bound Interval Skyline (BIS) method, Constrained Interval Skyline (CIS) algorithm	Interval skyline, constrained interval skyline	Interval uncertain data	Synthetic	Not realistic since not all range values can be assumed to fall within the median point	
Jiang et al. (2012)	Probabilistic skyline	Bounding-pruning-refining framework, Bottom-up, Top-down	Probabilistic skyline	Discrete uncertainty	NBA, synthetic	Require additional computational cost due to intensive probability calculation	
Li et al. (2017)	Probabilistic skyline	Distributed Interval Skyline Query (DISQ), Enhanced Distributed Interval Skyline Query (e-DISQ)	Interval skyline	Interval uncertain data	Apartments, synthetic	Require additional computational cost due to intensive probability calculation	
Saad et al. (2014, 2018)	Probabilistic skyline model	Skyline Query on Uncertain Dimension (SkyQUD) Framework	Skyline	Uncertain dimension (continuous real range)	NBA, synthetic	Rigid assumptions where users are only interested in either the minimum or maximum value over the whole space of a given dimension	
Saad et al. (2016, 2019)	Probabilistic skyline model	Skyline Query on Uncertain Dimension with Thresholding (SkyQUD-T) Framework	Range query	Uncertain dimension (continuous real range)	NBA, synthetic	Require additional computational cost due to intensive probability calculation	
Our work	X-Tree and Fuzzification	Constrained Skyline Queries over Uncertain Data (CSQUiD) Framework	Constrained skyline	Continuous range	NBA, synthetic	–	

Preliminaries

This section explains the concepts that are related to the work presented in this article. It also defines the terms and introduces the notations used throughout this article. Table 2 provides examples of objects with uncertain data, while Fig. 3 gives a pictorial representation of the six MBRs labelled as R1, R2, R3, R4, R5, and R6 that are constructed based on the given sample data. Note that the construction of the X-tree and its MBRs is omitted here, as interested reader may refer to Lawal et al. (2020a) for further details.

Table 2 Running example of objects with uncertain data.

Apartments	Rent rate ($)	Distance	
A	110–120	50	
B	120	25	
C	60–83	45	
D	78	105	
E	100	50	
F	120–160	40	
G	145	40	
H	85	45	
I	120	40	
J	160	45	

Figure 3 Graphical representation of MBRs for the sample data set provided in Table 2.

Based on the notations used in this article, we first provide the general definitions (i.e., Definition 1 through Definition 6) that have been defined either formally or informally in the literature (Alwan et al., 2016; Borzsonyi, Kossmann & Stocker, 2001; Lawal et al., 2020b; Saad et al., 2019). Then, we present the specific definitions that are relevant to our work in this article. These definitions assume a database D with m dimensions d={d1,d2,...,dm} and n objects O={o1,o2,...,on}.

Definition 1 Dominance Relationship: Object oi∈D is said to dominate object oj∈D where i≠j denoted as oi≺oj if and only if ∀dk∈d, oi.dk≤oj.dk∧ ∃dl∈d, oi.dl<oj.dl. Without loss of generality, we assume minimum value is preferred for all the dimensions. For example, referring to Table 2, apartment B is said to dominate apartment G since the values of apartment B for both dimensions, Rent Rate and Distance, are lesser than that of apartment G.

Definition 2 Exact Value: A value vi E R where R is a set of real numbers is said to be an exact value as its precise value is known/given.

Definition 3 Continuous Range Value: A value vi with a lower bound value, lb, and an upper bound value, ub, is a continuous range value as its precise/exact value is not known/given. In this work, a continuous range value is denoted as vi=[lb−ub] while the exact value of vi is a value that falls between both bounds including the endpoints. An example of a continuous range value is 110–120 which represents the rent rate of apartment A while the value 115 is one of the possible exact values of the given range.

Definition 4 Comparable Objects: Objects oi∈D and oj∈D where i≠j are said to be comparable if and only if ∀dk∈d, both oi.dk and oj.dk are in the form of exact value as defined by Definition 2. Otherwise, the objects oi and oj are said to be incomparable objects. For example, given the apartments G = (145, 40) and F = ([120–160], 40) as shown in Table 2, G and F are said to be incomparable objects, since the values of G.d1 and F.d1 are not comparable, as F.d1 = [120–160] is of the form of [lb−ub].

Definition 5 Constrained Query: A constrained query, cqi, over dk∈d is specified as [lb−ub] where lb and ub are the lower bound and upper bound values that defined the permissible range of values for dk of the given cqi.

Definition 6 Constrained Skyline Query: A constrained skyline query, cqi, retrieves the set of objects in D that lie within the constrained query [lb−ub], say D′whereD′⊆D, that are not being dominated by any other objects in D′. This is formally written as {oi|oi∈D′∧oj∈D′,oj≺oi}.

Definition 7 Left Vertex: Given an X-tree with a set of MBRs denoted as T={MBR1,MBR2,…,MBRz} with a search space Sp(x,y) defined as the root of the tree, where x≥0 and y≥0, each MBRw∈T contains four vertices denoted by {w.bl,w.br,w.tr,w.tl}. The vertex w.bl is referred to as the left vertex of the MBRw, as depicted in Fig. 4. The notation w.bl[x] is used to refer to the value of the x dimension of the vertex bl of MBRw. Based on the example given in Fig. 5 where MBRw={(1,2),(12,2),(12,9),(1,9)}, the left vertex of MBRw is given by (1, 2).

Figure 4 The vertices of an MBR.

Figure 5 The leftmost vertex of an MBR.

Definition 8 Leftmost Vertex: Given an X-tree with a set of MBRs denoted as T={MBR1,MBR2,…,MBRz} and a list of left vertices LV={LVMBR1,LVMBR2,…,LVMBRz}, LVMBRk∈LV is said to be the leftmost vertex of LV if and only if the distance between LVMBRk and Sp is the shortest as compared to the distances between other left vertices and Sp. Meanwhile, the notation MBRLMV is used to denote the MBR having the leftmost vertex.

Given four MBRs labelled as r={(10,39),(10,45),(15.5,45),(15.5,39)}, s={(16,35),(16,43),(23,43),(23,35)}, t={(7,28),(7,36),(14,36),(14,28)}, and u={(16.5,28),(16.5,36),(24,36),(24,28)}, while a constrained skyline query, cqb, defined on dimension x with constraint [5–25] as depicted in Fig. 6. The left vertices of r, s, t, and u are LVMBRr=(10,39), LVMBRs=(16,35), LVMBRt=(7,28), and LVMBRu=(16.5,28), respectively. By employing the Euclidean distance to calculate the distance between the left vertices of the MBRs and the Sp, the LVMBRt is identified as the leftmost vertex that lies within the given cqb.

Figure 6 The leftmost vertex among MBRs.

Definition 9 Dominance Relationship between MBRs: An MBRi={i.bl,i.br,i.tr,i.tl} is said to dominate an MBRj={j.bl,j.br,j.tr,j.tl} where i≠j denoted as MBRi≺MBRj if and only if the left vertex of MBRi < left vertex of MBRj, i.e., i.bl[x]<j.bl[x] and i.bl[y]<j.bl[y]. Obviously, object at the left vertex of an MBR is the dominant object of the MBR. For instance, the MBRt={(7,28),(7,36),(14,36),(14,28)} is said to dominate the MBRr={(10,39),(10,45),(15.5,45),(15.5,39)} since t.bl[x]=7<r.bl[x]=10 and t.bl[y]=28<r.bl[y]=39. The dominant object of r at (10, 39) will apparently be dominated by the dominant object of t at (7, 28). In this case t is recognized as the dominant MBR.

Definition 10 Associations between MBRs: The associations between MBRs are determined by analyzing the overlapping area (if any) between these MBRs. There are three cases that could occur, namely: non-overlapping, subset, and intersection, which are explained in the following:

Case I: Non-overlapping between MBRs–Given two MBRs, MBRr={r.bl,r.br,r.tr,r.tl} and MBRs={s.bl,s.br,s.tr,s.tl} where r≠s, MBRr and MBRs are said to be non-overlapping if and only if any of the following conditions hold: i) r.tl[x]≥s.br[x] or s.tl[x]≥r.br[x]

ii) r.tl[y]≤s.br[y] or s.tl[y]≤r.br[y]

Some examples of non-overlapping associations between MBRs are given in Figs. 7A and 7B.

Figure 7 Non-overlapping between MBRs.

Case II: Subset between MBRs-Given two MBRs, MBRr={r.bl,r.br,r.tr,r.tl} and MBRs={s.bl,s.br,s.tr,s.tl} where r≠s, MBRs is said to be a subset of MBRr denoted by MBRs⊆MBRr if and only if the following conditions hold: s.br[x]≤r.br[x], s.tl[x]≥r.tl[x], s.br[y]≥r.br[y], and s.tl[y]≤r.tl[y]. Some examples are given in Figs. 8A and 8B.

Figure 8 Subset between MBRs.

Case III: Intersection between MBRs–Given two MBRs, MBRr={r.bl,r.br,r.tr,r.tl} and MBRs={s.bl,s.br,s.tr,s.tl} where r≠s, MBRr and MBRs are said to intersect if and only if the following conditions hold: i) s.br[x]≥r.tl[x], s.br[y]≤r.tl[y], s.bl[x]≤r.br[x], and s.tl[y]≥r.br[y]

ii) MBRs is not a subset of MBRr as defined by Case II of Definition 10.

Some examples are provided in Figs. 9A–9D.

Figure 9 Intersection between MBRs.

Definition 11 Associations between MBRs and a Constrained Skyline Query cqi–There are four possible associations between MBRs and cqi that are lie within, intersection, overlap, and non-overlapping, as depicted in Figs. 10–13, respectively.

Figure 10 MBR lies within cqi.

Figure 11 (A and B) Intersection between MBR and cqi.

Figure 12 (A–C) Overlap between MBR and cqi.

Figure 13 Non-overlapping between MBR and cqi.

Case I: MBR lies within cqi–Given a constrained skyline query, cqi=[lb−ub] over dimension x, an MBRw={w.bl,w.br,w.tr,w.tl} is said to lie within the cqi if and only if the following conditions hold: w.tl[x]≥cqi.lb and w.br[x]≤cqi.ub. This is depicted by an example provided in Fig. 10.

Case II: Intersection between MBR and cqi–Given a constrained skyline query, cqi=[lb−ub] over dimension x, an MBRw={w.bl,w.br,w.tr,w.tl} is said to intersect the cqi if and only if any of the following conditions hold: i) w.tl[x]<cqi.lb and w.br[x]<cqi.ub

ii) w.tl[x]>cqi.lb and w.br[x]>cqi.ub

This is depicted by examples provided in Figs. 11A and 11B.

For each of the above cases, a new MBR is derived denoted as MBR w′ with a new set of vertices as follows: i) MBR w′={w.bl=(cqi.lb,w.bl[y]),w.br,w.tr,w.tl=(cqi.lb,w.tl[y])}

ii) MBR w′={w.bl,w.br=(cqi.ub,w.br[y]),w.tr=(cqi.ub,w.tr[y]),w.tl}

Case III: Overlap between MBR and cqi–Given a constrained skyline query, cqi=[lb−ub] over dimension x, an MBRw={w.bl,w.br,w.tr,w.tl} is said to overlap the cqi if and only if any of the following conditions hold: i) w.tl[x]<cqi.lb and w.br[x]=cqi.ub

ii) w.tl[x]<cqi.lb and w.br[x]>cqi.ub

iii) w.tl[x]=cqi.lb and w.br[x]>cqi.ub

This is typified by the examples provided in Figs. 12A–12C. For each of the above cases, a new MBR is derived and denoted as MBR w′ with a new set of vertices as follows: i) MBR w′={w.bl=(cqi.lb,w.bl[y]),w.br,w.tr,w.tl=(cqi.lb,w.tl[y])}

ii) MBR w′={w.bl=(cqi.lb,w.bl[y]),w.br=(cqi.ub,w.br[y]),w.tr=(cqi.ub,w.tr[y]),w.tl=(cqi.lb,w.tl[y])}

iii) MBR w′={w.bl,w.br=(cqi.ub,w.br[y]),w.tr=(cqi.ub,w.tr[y]),w.tl}

Note that MBR w′ is said to lie within the cqi, as defined by Case I of Definition 11.

Case IV: Non-overlapping between MBR and cqi–Given a constrained skyline query, cqi=[lb−ub] over dimension x, an MBRw={w.bl,w.br,w.tr,w.tl} is said to be non-overlapping the cqi if and only if the following conditions hold: i) w.tl[x]>cqi.ub or

ii) w.br[x]<cqi.lb

This is typified by the examples provided in Fig. 13.

The proposed framework

The Constrained Skyline Query processing on Uncertain Data (CSQUiD) is a framework that utilizes the MBRs of the X-tree indexing structure (Berchtold, Keim & Kriegel, 1996) that are constructed based on a given collection of uncertain data, to efficiently compute skylines of the constrained skyline queries. Instead of evaluating the objects that lie within a given constrained skyline query, cqi, only the objects of dominant MBRs are analyzed for deriving the dominant objects. The CSQUiD framework depicted in Fig. 14, consists of two distinct phases, namely: Data Pre-processing & Local Skylines Derivation (DP&LSD) and Fuzzification & Final Skylines Derivation (F&FSD). These phases are further explained in the following sections. Meanwhile, Algorithm 1 (Fig. 15) presents the general steps followed by CSQUiD to realize the dominant objects, S, from an uncertain database, Du, for a given cqi.

Figure 14 The CSQUiD framework (A) DP&LSD phase (B) F&FSD phase.

Figure 15 Algorithm 1 of CSQUiD.

Data pre-processing & local skylines derivation (DP&LSD)

At the DP&LSD phase, an X-tree indexing structure which consists of MBRs is constructed based on the given uncertain database, Du. These MBRs are leverage upon by the framework to streamline the processing of deriving skyline objects of a given constrained skyline query, cqi. As a result, the number of pairwise comparisons among objects is reduced. However, how to identify the MBRs that satisfy the conditions of the constrained skyline query needs to be explicitly specified. In this respect, several methods are employed by CSQUiD at this phase, namely: Cropping, Culling, and Grouping. The Cropping method is utilized to filter the MBRs of the X-tree that are relevant to a given constrained skyline query (see cases I, II, and III of Definition 11). MBRs that intersect, overlaps, or lie within the range defined by the constrained skyline query are identified and saved into a list named ND. Subsequently, the Culling method is utilized to get the MBR with the leftmost vertex, MBRLMV, by computing the distance between the Sp and the left vertices of the MBRs in the ND list which is then saved into the CD list. The Grouping method groups the objects in MBRLMV into two distinct groups that are Oc having objects with certain values while CS having objects with uncertain values. Then, the local skylines for the Oc group of MBRLMV are derived by employing the conventional skyline algorithm. By combining together the local skylines derived for Oc group with the CS group of MBRLMV, we realize the LS list which serves as input to the Fuzzification & Final Skylines Derivation (F&FSD) phase. The methods stated above are further elaborated in the following paragraphs.

Cropping Method: The Cropping method is employed after constructing an X-tree of a given uncertain database, Du, to identify the MBRs that either lie within, overlap, non-overlap, and intersect (see cases I, II, III, and IV of Definition 11) with the range specified by the constrained skyline query, cqi. Purposefully, this method is introduced to collect a set of MBRs into a list called ND whose objects would most likely contribute to the final skylines. The detail steps are as shown in Algorithm 2 (Fig. 16) with time complexity m∗O(n) where m is the number of MBRs of a given tree, T, and n=7 for the seven different cases as presented in the algorithm. By applying Algorithm 2 on the example provided in Fig. 17, with cqi=[55−120], the MBRR4, MBRR5, and MBRR6, are returned as the MBRs that satisfy the conditions stated in Definition 11. Here, MBRR4 is said to overlap with the cqi, while MBRR5 and MBRR6 are said to intersect the cqi.

Figure 16 Algorithm 2 of CSQUiD.

Figure 17 MBR with the leftmost vertex.

Culling Method: The Culling method is employed to identify the MBR with the leftmost vertex, MBRLMV, by computing the Euclidean distance of the left vertices of MBRs realized in ND to Sp which is the left vertex to the root of the X-tree. Instead of evaluating the dominance relationship between objects of MBRs in the ND, only the set of objects within the MBR with the leftmost vertex, MBRLMV, and those objects of MBRs not dominated by MBRLMV are processed in identifying the dominant objects for the given constrained skyline query, cqi. Essentially, the number of pairwise comparisons is reduced since many unnecessary comparisons among objects are avoided. This is due to the fact that the MBRs dominated by MBRLMV are pruned off. The MBRLMV and the dominant MBRs are then passed to the Grouping method. The detail steps of the Culling method are as delineated in Algorithm 3 (Fig. 18).

Figure 18 Algorithm 3 of CSQUiD.

Using the example depicted in Fig. 17, with cqi=[55−120], the left vertices for the MBRs in ND of the given example are MBRR4=(100,50), MBRR5=(60,45), and MBRR6=(120,25) with distances computed to Sp(0,0) as 47.16, 22.36, and 60, respectively. With this, the MBR with the leftmost vertex is MBRR5. Nonetheless, the objects of MBRR5 and MBRR6 are collected since MBRR5 does not dominate MBRR6; while it dominates MBRR4. Thus, objects {B, C, D, H, I} are the objects passed to the next method, Grouping method.

Grouping Method: Based on the CD derived by the previous method, the Grouping method is employed to group the objects into two distinct groups. The first group, Oc, consists of objects with certain data, while the second group, CS, consists of objects with uncertain data. The detail steps of the Grouping method are as demonstrated in Algorithm 4 (Fig. 19). The result of deploying the Grouping method is presented in Fig. 20.

Figure 19 Algorithm 4 of CSQUiD.

Figure 20 Grouping objects of MBR.

Conventional Skyline Algorithm: To determine the local skylines, the Oc group of objects is subjected to the conventional skyline algorithm. {B, D, H} is the set of local skylines of Oc in Fig. 20, based on the instances shown in Fig. 17. In a single list, LS, the items of Oc and CS are handed to the subsequent phase.

Fuzzification & final skylines derivation (F&FSD)

The Trapezoid Fuzzification method is used at the Fuzzification & Final Skylines Derivation (F&FSD) phase to generate the trapezoid membership function values for each continuous range value of the objects in LS, and the Sum Aggregation method is used to return a single trapezoid membership function value for the values derived by the Trapezoid Fuzzification method. The final skylines of the constrained skyline query, cqi, are computed using the conventional skyline algorithm, while the Centroid Defuzzification method is employed to return the exact value to the continuous range value. These techniques are explained in more detail below.

Trapezoid Fuzzification Method–This inductive approach typically creates a fuzzy set by assigning a trapezoid membership function to a set of independent observations (Chiu, 1996; Coffman-Wolph, 2016). The CSQUiD framework generates a trapezoid fuzzy set by first normalising a continuous range value of an object into segments. Next, the kNN algorithm is applied to identify the k objects with either an exact value or a midpoint value that are closest to the given continuous range value. The Trapezoid Fuzzification method then determines and uses the exact value or midpoint value that has the shortest distance to each segment to derive its trapezoid membership function values. Given a segment of a continuous range value, uj=[uj.lb−uj.ub], its midpoint, cpuj, is computed as cpuj=(uj.ub−uj.lb)/2 where uj.lb and uj.ub are the lower bound and upper bound values of uj, respectively. With the cpuj value, a trapezoid fuzzy set of the segment is derived; denoted by fs(cpuj,mfvuj). Meanwhile, mfvuj={mfvu1, mfvu2,…, mfvun} represents the trapezoid membership function values of all the segments of the given continuous range value, vi. The degree to which a continuous range value belongs to a fuzzy set is represented by the trapezoid membership function value, which is a real continuous interval [0,1]. The endpoint of 0 indicates no membership and 1 indicates full membership, while values between the endpoints represent different degrees of membership. Interested readers may refer to (Chiu, 1996; Coffman-Wolph, 2016; Ross, 2000; Zadeh, 1996), to get further details on the Trapezoid Fuzzification method.

As an example, consider the apartment C with the continuous range value [60 − 83] presented in Fig. 17. To determine the set of segments, the midpoint is calculated as [(83 + 60)/2] = 71.5, while the unit scale of each segment is derived through the normalization of the range specified by the continuous range value C as [(71.5 − 60)/(83 − 60)] = 0.5. Thus, using the derived segment unit scale, the set of segments of [60 − 83] is represented as {[60−60.5],[60.5−61],...,[82.5−83]}. To compute the trapezoid membership function value for the derived segments, the midpoint of each segment is calculated. The segments within the range [60 − 83], having the shortest distance between cpuj and D(78) are [77.5 − 78] and [78 − 78.5], while the midpoint for these segments are 77.75 and 78.25, respectively. The trapezoid membership function values for the continuous range value [60 − 83] for apartment C have 0.55694799, 0.49839744, and 0.50160256 as the shortest distances to the exact value of D(78).

Sum Aggregation Function–The Sum Aggregation function is used to get a single trapezoid membership function value to the continuous range value. The single value for the trapezoid function values generated for the closest segments to object D is computed by adding 0.55694799 + 0.49839744 + 0.50160256 = 1.55694799. Based on the sum aggregation method, max (1,1.55694799) = 1.55694799 is realized as the sum aggregate value for the trapezoid membership function values of [60 − 83].

Centroid Defuzzification Method: The Centroid Defuzzification method returns the computed exact value for the single trapezoid membership function value. The result is found along the range defined by a continuous range value. For example, 71.5 + 1.55694799 = 73.05694799 is the exact value for the continuous range values [60 − 83] of apartment C. Figure 21 shows the data set Du with objects having exact values.

Figure 21 Du with objects having exact values.

Results and discussions

In this section, through an extensive experimental analysis conducted on both synthetic and the NBA real data set, the experimental results for processing constrained skyline queries over uncertain data were obtained. With a constrained skyline query cqi=[lb−ub] defined on uncertain data, the performance result of the proposed CSQUiD framework with respect to the CIS and SkyQUD-T algorithms by (Li et al., 2012) and (Saad et al., 2019), respectively, were discussed and analyzed.

Experimental settings

The performances of CIS, SkyQUD-T and CSQUiD algorithms are measured based on the CPU processing time and the number of pairwise comparisons. The experiments make use of two data sets, namely: synthetic and real data sets. The synthetic data set includes correlated, anti-correlated, and independent data sets. The experimental parameters used in the performance analysis include size of the data set (n), data distribution ( σ), number of dimensions (d), and the size of the constrained skyline query ( λ) as presented in Table 3. The method utilized in the work of Saad et al. (2019) is employed for uniformly generating the size of a constrained skyline query ( λ) as a percentage of the volume of the data sets. For each size, for instance 0.1%, 50% constrained skyline queries are generated at random and the average results obtained are reported as the final results.

Table 3 Experimental parameter settings.

Parameter	Values	
Data set type	Synthetic	NBA	
Size of data set ( n)	100 k	21,961	
Data distribution ( σ) (%)	50	
Number of dimension ( d)	3	17	
Size of constrained query ( λ) (%)	0.1, 8, 16, 32, 64, 98, 98.5, 99, 99.5	

Synthetic data set

The synthetic data sets which include anti-correlated, independent, and correlated as illustrated in Fig. 22 are generated using the same generator as used by Lawal et al. (2020a) and Saad et al. (2016). Every data set contains 100 k objects, each of which has three dimensions that corresponds to a uniform random variable with values between 1 and 100. Further, to ensure that the distribution between exact values and continuous range values is 50%, we set one of each object’s dimensions to represent the uncertain data in the form of a continuous range value with a length between 1 and 100.

Figure 22 An illustration of the synthetic data set (Lawal et al., 2020a).

Real data set

The National Basketball Association (NBA) statistic (www.basketballreference.com), a real data set that represents various statistic values associated with NBA players, is also used in the performance analysis of this study. It has been widely employed in other studies that focus on skyline queries over uncertain data (Lawal et al., 2020a; Saad et al., 2016, 2018, 2019; Tan, Eng & Ooi, 2001). The NBA data set contains 21,961 objects with a total of 16 dimensions. Since the NBA data set is a complete data set with exact values, a new dimension representing the uncertain dimension is added to the data set. The values of this dimension are in the form of a continuous range that are randomly generated using the same procedure that was employed to generate the synthetic data set.

Eperimental results

We present the CIS, SkyQUD-T, and CSQUiD performance analyses in supporting constrained skyline queries below. The query size is adjusted during the analysis, ranging from 0.1% to 99.5%. This analysis’s goal is to confirm how well these methods work with queries of varying widths, ranging from the smallest range (0.1%) to the highest range (99.5%). The analysis’s findings for CPU processing time and number of pairwise comparisons are shown in Figs. 23 and 24, respectively.

Figure 23 (A–D) The CPU processing time of CIS algorithm, SkyQUD-T and CSQUiD frameworks by varying the size of constrained queries.

Figure 24 (A–D) The number of pairwise comparisons of CIS algorithm, SkyQUD-T and CSQUiD frameworks by varying the size of constrained queries.

From Fig. 23, the following can be observed: (i) In most data sets, a slight increment in the CPU processing time can be seen in all solutions, when the size of the constrained skyline query is increased. Intuitively, the larger the size of the query, the more spaces it covers and consequently more objects need to be analyzed. (ii) For anti-correlated data set, the CPU processing time for all solutions is relatively the same as can be seen in Fig. 23A. Nonetheless, CSQUiD shows a slight better performance compared to CIS and SkyQUD-T. (iii) For correlated and independent data sets, the performance of both SkyQUD-T and CSQUiD is by far better than that of CIS, even though there is an exception at 8% of the constrained skyline query, as presented in Figs. 23B and 23C, respectively. (iv) For the real data set, NBA, CSQUiD shows better performance while both SkyQUD-T and CIS have similar performance as clearly shown in Fig. 23D.

In most cases CSQUiD gained better performance with regard to the CPU processing time. This is due to several reasons, as discussed in the following: CSQUiD employed the X-tree indexing technique to organize the objects into MBRs. Objects having similar features are grouped into the same MBRs. Given a constrained skyline query, the MBRs that satisfy the constraints of the query are identified. Only the objects of the dominant MBRs are analyzed further while those objects of the dominated MBRs (although initially they satisfy the constraint of the query) are discarded from skyline computation. Utilizing the fuzzification approach, the exact value of each continuous range value is predicted; which then enable the conventional skyline algorithm to be applied to derive the skylines. Similar to CSQUiD, CIS employed the R*-tree to organize the objects of the uncertain data set. Unlike CSQUiD, the MBRs that satisfy the constraints of the query are identified and the objects of these MBRs are analyzed. While, the median approach is employed to get the exact value of each continuous range value. This means that only objects of the dominants MBRs are analyzed by CSQUiD; while objects of those MBRs that are within the constraints of the query are analyzed by CIS. Obviously, the number of objects involved in the skyline computation of CSQUiD is lesser than the CIS algorithm as unnecessary objects are filtered in the earlier stages. Meanwhile, SkyQUD-T requires scanning every object in the data set to identify the relevant objects, i.e., those objects that are within the constraints of the given query; before extensive probabilistic calculation and thresholding are performed between every pair of these relevant objects. Nonetheless, when the size of the constrained skyline query is relatively small (0.1–8%), the chances of an MBR to dominate the other MBRs are low. Consequently, the performance of CSQUiD, CIS, and SkyQUD-T with regard to CPU processing time for such case is almost similar.

From Fig. 24, the following can be observed: (i) In most data sets, a slight increment in the number of pairwise comparisons can be seen in all solutions, when the size of the constrained skyline query is increased. This is mainly due to the fact that a larger size of query will span a larger space and consequently covers more objects. (ii) For anti-correlated and correlated data sets, the number of pairwise comparisons of SkyQUD-T is the highest while both the CSQUiD and CIS show almost similar performance as presented in Figs. 24A and 24B, respectively. (iii) For independent and NBA data sets, the performance of CSQUiD is by far better than that of CIS and SkyQUD-T, as presented in Figs. 24C and 24D, respectively.

Given a data set with m dimensions and n objects, the average number of pairwise comparisons is m[(n(n−1))/2]. The objects analyzed by CSQUiD are those of the dominant MBRs that satisfy the constraint of the constrained skyline query. While the objects analyzed by CIS are the objects of all the MBRs that satisfy the constraint of the constrained skyline query. Obviously, the number of pairwise comparisons performed by CSQUiD is lesser than CIS. Meanwhile, the objects analyzed by SkyQUD-T are those that satisfy the constraint of the constrained skyline query. The pairwise comparisons performed by both CSQUiD and CIS are based on the precise values (either the initial value or predicted/median value) while the pairwise comparisons performed by SkyQUD-T are based on the initial value, which can either be exact or continuous range value. Nonetheless, when the size of the constrained query is relatively small (0.1–8%), the performance of CSQUiD, CIS, and SkyQUD-T with regard to number of pairwise comparisons is almost similar. This is mainly because the number of MBRs that fall within the constraint of the given query is small and the chances of an MBR to dominate the other MBRs are low.

From the experimental results presented in Figs. 23 and 24, it is obvious that CSQUiD is superior and outperforms other algorithms in terms of CPU processing time and number of pairwise comparisons. Hence, employing an indexing technique like X-tree, applying the dominance relationship between MBRs, avoiding extensive probability computation and thresholding, and predicting the exact values of continuous range values have enhanced the performance of the CSQUiD in processing constrained skyline queries.

Table 4 presents the percentages of improvement gained by CSQUiD as compared to CIS and SkyQUD-T algorithms. With regard to the number of pairwise comparisons, CSQUiD outperforms CIS and SkyQUD-T algorithms by 30.32–34.60%, 5.85–58.42%, 47.13–60.15%, and 79.71–88.68% for anti-correlated, correlated, independent, and NBA data sets, respectively. Meanwhile the percentages of improvement gained with regard to CPU processing time are as follows: 12.09–19.37%, 16.17–21.92%, 21.52– 37.06%, and 24.68–30.31%, for anti-correlated, correlated, independent, and NBA data sets, respectively.

Table 4 Percentage of improvement of CSQUiD framework by varying the size of constrained queries.

Algorithm compared	Data set	Number of pairwise comparisons	CPU processing time	
CIS	Anticorrelated	34.60%	19.37%	
	Correlated	5.85%	21.92%	
	Independent	47.13%	37.06%	
	NBA	88.68%	30.31%	
SkyQUD-T	Anticorrelated	30.32%	12.09%	
	Correlated	58.42%	16.17%	
	Independent	60.15%	21.52%	
	NBA	79.71%	24.68%	

Conclusion

The skyline process is considered expensive due to the exhaustive domination tests performed to identify the skylines. Skyline search space and computation process on uncertain data are affected by parameters such as the number of dimensions, data set size, and the size of the constrained query. In this article, an efficient framework called CSQUiD is proposed to address the problem associated with processing constrained skyline queries over uncertain data. This framework consists of two phases, namely: Data Pre-processing & Local Skylines Derivation and Fuzzification & Final Skylines Derivation. The DP&LSD phase utilizes an X-tree indexing technique to organize the uncertain database into MBRs; where objects with similar uncertainty values are organized into the same MBR. The Cropping, Culling, Grouping methods besides the conventional skyline algorithm are invoked in this phase to derive the local skylines of each identified dominant MBR. Meanwhile, F&FSD phase adopts the Trapezoid Fuzzification, Sum Aggregation, and Centroid Defuzzification methods to predict an exact value to a continuous range value before the conventional skyline algorithm can be applied to derive the final skylines. Various sets of experiments have been accomplished to prove the efficiency and effectiveness of the CSQUiD framework over the recent existing frameworks that are CIS and SkyQUD-T. The results have proven that our proposed solution is outclassing all the existing solutions in computing constrained skyline queries over uncertain data.

There are several enhancements that can be made based on the findings presented in the article. These include (I) Processing multiple constrained skyline queries as a batch–the CSQUiD framework handles a constrained skyline query at a time. Multiple constrained skyline queries can be evaluated as a batch and further improvement can be achieved by analyzing the similarities between these queries with regard to the constraints specified in each query. Similar constraints can be grouped and evaluated together instead of evaluating each query separately. This will avoid unnecessary skyline computation. (II) Processing constrained skyline queries over data stream–handling constrained skyline queries over data streams is a challenging task as these streams of data are known to have the properties of time varying (time sensitive), continuous, real time, volatile, and unrepeatable. By utilizing the sliding window approach, the CSQUiD framework can be employed to dynamically update the skyline objects at each window. (III) Maximizing user’s preference function–to ensure users are given flexibility to specify their interest as their query requirements, different forms of skyline retrieval like subspace skyline, top-k skyline, k-dominate skyline, and k representative skyline can be explored. For example, subspace skyline allows users to define their preferences on various subsets of dimensions. It is also effective in protecting the privacy and anonymity of data, with only specific dimensions being accessible. In the meantime, top-k skyline filters the best k skylines, where k is the number of answers the user desires, rather than returning all skyline objects, which are probably large in size. Additionally, the k representative skyline provides an intuitive way to identify the k most significant objects that can represent the corresponding full skyline. Exploring these types of skyline queries may also address the shortcomings of the CSQUiD framework, as mentioned earlier.

Supplemental Information

Supplemental Information 1 Code.

Supplemental Information 2 Results.

Supplemental Information 3 Data from NBA.

Additional Information and Declarations

Competing Interests

Author Contributions

Data Availability

The authors declare that we have no known competing financial interests or personal relationships that could have appeared to influence the work reported in this article.

Ma'aruf Mohammed Lawal conceived and designed the experiments, performed the experiments, analyzed the data, performed the computation work, prepared figures and/or tables, authored or reviewed drafts of the article, and approved the final draft.

Hamidah Ibrahim conceived and designed the experiments, authored or reviewed drafts of the article, and approved the final draft.

Nor Fazlida Mohd Sani performed the experiments, authored or reviewed drafts of the article, and approved the final draft.

Razali Yaakob performed the experiments, authored or reviewed drafts of the article, and approved the final draft.

Ali A. Alwan performed the experiments, authored or reviewed drafts of the article, and approved the final draft.

The following information was supplied regarding data availability:

The statistical data set of players obtained from National Basketball Association are available in the Supplemental File.

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
