# Peer review of "CSQUiD: an index and non-probability framework for constrained skyline query processing over uncertain data"

_PeerJ Computer Science, doi:10.7717/peerj-cs.2225_

## Round 0.1 · original submission · Major Revisions

Please consider the comments from the reviewers.

**Language Note:** The review process has identified that the English language must be improved. PeerJ can provide language editing services - please contact us at [email protected] for pricing (be sure to provide your manuscript number and title). Alternatively, you should make your own arrangements to improve the language quality and provide details in your response letter. – PeerJ Staff

Reviewer 1 ·

Basic reporting

1. The figures are not clear. High-resolution figures should be included.
2. In Table 2, the authors must include the limitations of existing schemes. Which motivates the authors to do this research.
3. In Algorithm 2, too many else-if statements are used. Which can lead the algorithm to high complexity. It seems the algorithm is with high complexity or should derive the complexity of the algorithms.

Experimental design

Fine.

Validity of the findings

Valid and Accurate.

Additional comments

N/A

Reviewer 2 ·

Basic reporting

The document structure is hard to follow.
The references list is extensive and doesn't focus on a subject.
The presentation is dense and difficult to read.

Experimental design

The use of appendixes would be constructive.
The presentation should be restructured.
Hard to read.

Validity of the findings

The results seem to be reliable.

Additional comments

My suggestion is to redraft the entire document if it has to be accepted

---

## Round 0.2 · accepted · Accept

The comments for the Reviewer 2 were accurately revised. The manuscript can be accepted.

Reviewer 1 ·

Basic reporting

The authors incorporated the comments accordingly.

Experimental design

N/A

Validity of the findings

N/A

Additional comments

N/A